# A Longitudinal Study of Hematology and Stress Biomarker Profiles in Young Asian Elephants (*Elephas Maximus*) in Relation to Elephant Endotheliotropic Herpesvirus (EEHV) in Thailand

**DOI:** 10.3390/ani11092530

**Published:** 2021-08-28

**Authors:** Khajohnpat Boonprasert, Yaoprapa Yun, Worapong Kosaruk, Patcharapa Towiboon, Pallop Tankaew, Veerasak Punyapornwithaya, Thittaya Janyamathakul, Panida Muanghong, Janine L. Brown, Chatchote Thitaram, Chaleamchat Somgird

**Affiliations:** 1Center of Elephant and Wildlife Research, Faculty of Veterinary Medicine, Chiang Mai University, Chiang Mai 50100, Thailand; khajohnpat@gmail.com (K.B.); yaoprapam@gmail.com (Y.Y.); woraph.kosa@gmail.com (W.K.); towiboon@gmail.com (P.T.); pallop_off@hotmail.com (P.T.); BrownJan@si.edu (J.L.B.); cthitaram@gmail.com (C.T.); 2Department of Food Animal Clinic, Faculty of Veterinary Medicine, Chiang Mai University, Chiang Mai 50100, Thailand; pveerasak.r@gmail.com; 3Veterinary Public Health Centre and Food Safety for Asia Pacific (VPHCAP), Faculty of Veterinary Medicine, Chiang Mai University, Chiang Mai 50100, Thailand; 4Pattara Elephant Farm, Chiang Mai 50100, Thailand; thittayavet68@gmail.com; 5Mae Taeng Elephant Park and Clinic, Mae Thang, Chiang Mai 50150, Thailand; panida.vet72@hotmail.com; 6Center for Species Survival, Smithsonian Conservation Biology Institute, Front Royal, VA 22630, USA; 7Department of Companion Animal and Wildlife Clinics, Faculty of Veterinary Medicine, Chiang Mai University, Chiang Mai 50100, Thailand

**Keywords:** Asian elephant, hematology parameters, stress indicators, immunoglobulin A, glucocorticoids, elephant endotheliotropic herpesvirus

## Abstract

**Simple Summary:**

A change in hematology profiles is one indicator of EEHV infection before clinical signs appear; however, to be effective, individual baselines and age-matched reference values are needed. A longitudinal investigation of viremia, hematology values, and stress biomarkers was performed in three non-EEHV and six prior infected EEHV calves to better understand EEHV-HD-associated factors. Blood, saliva, and feces were collected for 1 year for analysis of complete blood count (CBC), viral load, glucocorticoids (GCs), and Immunoglobulin A (IgA). Results did not differ between the groups, except for one elephant that presented with EEHV-HD during the study and exhibited high viremia, altered hematology profiles, and decreased stress biomarker concentrations. Thus, as in other studies, hematology changes were associated with EEHV infection, while preliminary data in one calf suggests that stress-response measures might also be informative and warrant further investigation.

**Abstract:**

Elephant endotheliotropic herpesvirus hemorrhagic disease (EEHV-HD) is a virulent disease that causes severe hemorrhage and sudden death in Asian elephant calves. A change in hematology profiles is one indicator of infection before clinical signs appear; however, to be effective, individual baselines and age-matched reference values are needed. Stress has been speculated to be a factor in clinical EEHV cases, but relationships have not been demonstrated empirically. This study evaluated blood hematology and several stress response markers—salivary cortisol, fecal glucocorticoid metabolites (FGM), salivary Immunoglobulin A (SIgA), and fecal IgA (FIgA) in samples collected for 1 year from three healthy calves with no EEHV history (non-EEHV), and six that had previously been infected, developed clinical signs and survived (prior-EEHV). Hematology values between non-EEHV and prior-EEHV elephants were not different and within published reference ranges. Concentrations of salivary cortisol, FGM, SIgA, and FIgA also were variable and showed seasonal differences, but no relationships to prior EEHV status. One of the prior EEHV calves became re-infected, developed hemorrhagic disease (HD), and died during the study period. That calf exhibited lymphocytopenia, monocytopenia, and thrombocytopenia. Additionally, all stress biomarker concentrations were lower in the 12 days before viremia was observed. Thus, as in other studies, changes in hematology occur with EEHV infection, while preliminary data in one calf suggests that stress-response measures might also be informative and should be studied further.

## 1. Introduction

Elephant endotheliotropic herpesvirus (EEHV) is a cause of the fatal hemorrhagic disease (HD) in Asian elephants worldwide and primarily affects calves [1,2]. Clinical signs include head and thoracic limb edema, lethargy, anorexia, and cyanosis of the tongue [1,3]. Death can occur quickly, within 1–7 days after clinical onset [4,5], with a mortality rate of up to 85% [1,6]. Identifying changes in blood hematology in response to EEHV infection are useful in diagnosing and monitoring disease progression [7], being detectable several days before clinical signs manifest [2,8,9]. Lymphopenia, monocytopenia, and thrombocytopenia are hallmarks of EEHV-HD viremia [2,8,10,11,12]. Thus, monitoring changes in white blood cell (WBC) counts is recognized as an important tool in the diagnosis of EEHV and assessing treatment efficacy in suspected cases [2,13].

To be most effective, a standard hematology profile is needed for each elephant to establish baseline values, as they can vary considerably [2,14]. Baseline values also are needed to detect often subtle changes after EEHV infection. Early detection of a drop in WBC compared to normal baseline can prove to be diagnostic and predictive for making rapid treatment decisions before clinical appearance [11,12], which improves treatment outcomes and survival [2]. To date, most hematology reference values in Asian elephants have been calculated on a population level [14,15,16], often with limited samples across age classes [2,11], and from elephants that differ in geography, nutrition, and management [17]. Therefore, establishing standard hematology values specific to clinically healthy young elephants could be useful for the assessment of suspected EEHV cases in this age category.

Similar to other viruses in the Herpesviridae family, EEHV can be symptomatic or asymptomatic [3,4,18,19,20,21,22]. Latent EEHV infections have been identified in the salivary gland and intestinal epithelial cell samples [23], with EEHV reactivation being hypothesized to be a result of an immunocompromised state, potentially related to stress [1]. For example, associations have been found between EEHV infections and weaning or training in the 30 days leading up to infection [24]. Physiological stress also has been implicated in increasing the susceptibility of young calves to EEHV infection and risk of developing EEHV-HD [18,25], although the mechanisms are not well understood.

Studies in zoo elephants have characterized cortisol patterns, finding higher concentrations in males than females due to management differences [14] and in cycling compared to non-cycling females [26]. Cortisol is a product of the hypothalamic-pituitary-adrenal axis, which is activated at times of physical or physiological stress [1,27,28,29]. As such, it is widely used as a biological indicator of the stress response and welfare state in many species [28], including Asian elephants [30,31,32,33,34,35]. However, data on how stressful events, such as transportation, pregnancy, weaning, husbandry, and management changes are related to clinical EEHV activation and cortisol secretion are lacking.

IgA is another potential biomarker of health and welfare [36] and is found in many secretory fluids, including saliva, breast milk, and nasal, gastrointestinal, bronchial, and urogenital secretions [37]. IgA is the first line of immune defense at the mucosal surfaces of the gastrointestinal, respiratory, and genitourinary tracts, where more than 95% of infections are initiated [38]. In animals, restraint and isolation have been associated with lower IgA [39,40], whereas higher concentrations were indicative of a positive welfare state [36,41,42].

While changes in hematology are known to be useful prognostic indicators of EEHV infection before clinical signs appear [2,4,12], clear links between EEHV status and stress biomarkers like glucocorticoids or IgA have not been established. This study explored links between viremia and hematology profiles, in addition to several stress-response measures—salivary cortisol (SC), FGM, salivary IgA (SIgA), and fecal IgA (FIgA) in Asian elephant calves. The goal was to improve our understanding of factors associated with EEHV viremia and provide more diagnostic tools to predict survival outcomes.

## 2. Materials and Methods

### 2.1. Elephants and Sample Collection

All animal procedures were approved by the Institutional Animal Care and Use Committee, Faculty of Veterinary Medicine, Chiang Mai University, Chiang Mai, Thailand (reference number S2/2562). Nine juvenile Asian calves from elephant camps in Chiang Mai, Thailand were included in this study, with an average age of 49.0 ± 7.1 months. Elephant calves were categorized as (1) ‘non-EEHV’, which consisted of three calves aged 56.0 ± 4.0 months (range 48–60 months) that did not have a history of EEHV infection based on medical records, and were not viremic based on real-time PCR in the month before study onset; and (2) ‘prior-EEHV’, comprised of six elephant calves aged 45.5 ± 10.5 months (range 20–78 months) that had previously tested positive for EEHV by blood PCR and presented clinical signs of EEHV-HD, but were not viremic in the month before study onset. One elephant in the second group tested positive again for EEHV, showed clinical signs, and died during the study period. Elephants received general health checks from a veterinarian before and during the study. Calves in the study had been weaned before study onset, with an average weaning age of 25.5 ± 3.8 months. Calves were fed Bana grass, sugarcane, banana, and banana tree. All lived in their natal camp, except one (E4) that was moved about 600 km from the northeastern to the northern region of Thailand at the age of 17 months (about 2 months after weaning). That animal was diagnosed with EEHV 11 months after arriving at the new facility and 7 months prior to this study. General information about the elephants in this study is summarized in Table 1.

### 2.2. Blood Collection

Blood samples were collected from an auricular vein once a month from December 2018–November 2019, after salivary and fecal collection to prevent increased cortisol concentrations due to restraint [1,29]. Blood was collected into EDTA anticoagulant tubes for hematology and EEHV real-time PCR analyses. Samples were chilled at 4 °C immediately after collection and transferred to the Veterinary Diagnosis Center, Faculty of Veterinary Medicine, Chiang Mai University for analysis.

### 2.3. Saliva and Fecal Analysis

Saliva and fecal samples were collected twice a month during the 2nd and 4th week between 10.00–12.00 h, to avoid variation in circadian rhythms of cortisol [33,43]. Saliva was collected using a Salivette^®^ kit (Sarstedt, AG&Co, Numbrecht, Germany) and harvested by centrifuging at 1500× *g* for 2 min at 15 °C [42]. Fecal balls were mixed and approximately 20 g was collected into two separate zip-lock bags. All samples were transported in a 2–8 °C container box [42] to the Faculty of Veterinary Medicine, Chiang Mai University. All samples were stored at −20 °C until analysis [42,44,45]. Fecal samples were extracted as described previously for GCs [34,42] and IgA [42,46].

### 2.4. Salivary Cortisol and FGM Analyses

Salivary cortisol concentrations were measured by a double-antibody enzyme immunoassay (EIA) using a secondary goat anti-rabbit IgG antibody and polyclonal rabbit anti-cortisol antibody (R4866, Coralie Munro, University of California Davis, Davis, CA, USA) validated for elephants [42,44].

### 2.5. SIgA and FIgA Analyses

IgA in salivary and fecal samples was measured using commercially available components as described by Edwards et al. [46] with some modifications as described by Plangsangmas et al. [44] and Kosaruk et al. [42].

### 2.6. EEHV Analysis

Whole blood samples were processed within 24 h after collection using a commercial kit (NucleoSpin^®^, Macherey-Nagel GmbH & Co. KG, Dueren, Germany) in accordance with the manufacturer’s recommended protocol for 200 μL of whole blood, and extracted DNA was eluted in 100 µL of elution buffer. Concentrations of DNA were calculated by use of a spectrophotometer [9].

All DNA samples were screened for EEHV1 and EEHV4 using quantitative real-time PCR. Primers for genes U38 (polymerase), U39 (glycoprotein B), and U40 (Terminase) specific to EEHV1 and EEHV4 were designed and synthesized. The quantification of EEHV DNA was performed in duplicate on ABI 7300 Real-Time PCR System equipped with SDS Software v1.4 (Life Technologies, Darmstadt, Germany).

RT-PCR was performed using SensiFAST SYBR Hi-ROX Kit (Bioline, London, UK) as per the manufacturer’s instruction. The cycling conditions were set as follows: an initial denaturation of 95 °C for 2 min, followed by 40 cycles of denaturation at 95 °C for 5 s, and annealing/extension at 60 °C for 30 s in every tested gene. Subsequently, specificity was confirmed by dissociation curve analysis (Tm). Correct product sizes were also determined by 2% agarose gel in 0.5× tris-acetate-ethylenediaminetetraacetic acid (TAE) buffer at 100 V for 35 min. The gels were stained with ethidium bromide (0.5 μg/mL) were used to check real-time PCR product specificity. Samples that gave a PCR product of the expected size with all primer pairs, as determined by gel electrophoresis, will be classified as EEHV1 or EEHV4 positive [47].

The EEHV1 and EEHV4 genome copy numbers measured during the quantitative real-time PCR process for EEHV and genomic TNFα were first normalized against the spiked internal control to compensate for differences in nucleic acid extraction efficiency for each blood sample. The EEHV1 and EEHV4 genome copies for each blood sample were then normalized against those of genomic TNFα performed in parallel with 1 µg of extracted DNA [48].

### 2.7. Statistical Analysis

R statistical software v3.5.1 was used to conduct all statistical analyses in this study. Data were averaged by season: summer (mid-February to mid-May); rainy (mid-May to mid-October); winter (mid-October to mid-February) followed by the Meteorological Department of Thailand (www.tmd.go.th, accessed on 12 July 2020). Normality and variance of the data were examined by QQ plot (R package: Quantile-Quantile plot 0.0.4; qqplotr) [49]. Mean data for hematology parameters and stress indicator concentrations are shown as a mean ± standard error of the mean (SEM). Hematology values and stress indicator concentrations from healthy elephants with no clinical signs of illness during the study period (all except E5) were used to calculate normal reference values. Repeated measures data were analyzed using the Generalized Least Squares method (GLS, R package: non-linear mixed effect model 3.1–148; nlme [50]) to determine the effects of infection status and season factors on hematology values and stress indicator concentrations. Tukey’s multiple comparisons of the least-squares mean (LS-mean) were used as post hoc tests to determine differences between variable categories. Unpaired two-sample *t*-tests were used to compare differences between the mean study mean for health and clinic EEHV-HD elephants.

## 3. Results

### 3.1. Animals

General health evaluations found eight of the calves were healthy during the study, with no clinical signs of illness observed. Only one elephant (E5, 27-months old) in the non-fatal EEHV group developed signs of EEHV-HD, e.g., diarrhea, depression, lethargy, and facial edema, and died within 72 h (Table 1). A summary of viral load, hematology values, and stress indicator concentrations of E5 throughout the study is shown in Table 2. This animal had survived an earlier infection with EEHV1A in June 2018 before study onset but was later infected with EEHV1A during the study in June 2019. During the first infection, the animal presented signs of lethargy, depression, and facial edema. This animal was treated with an antiviral drug (famciclovir, Famvir^®^, Novartis Co., Ltd., London, UK) per rectal immediately on the first day clinical signs were observed. On the 3rd day of treatment, clinical signs resolved, although antiviral treatment was continued for 7 days. However, during the second infection, the animal developed severe diarrhea for 2 consecutive days (14 and 15 June 2019) prior to the manifestation of EEHV-HD clinical signs on Day 3 (16 June 2019), and so was transferred to the elephant hospital in Lampang (National Elephant Institute, Forest Industry Organization) for intensive treatment, including administering a different antiviral drug (acyclovir, Vilerm^®^ IV Infusion, Siam Pharmaceutical Co., Ltd., Bangkok, Thailand), plasma therapy, and supportive fluids and multivitamins. This animal did not respond and died on Day 4 (19 June 2019) of treatment in the hospital.

### 3.2. Hematological Values

Hematology results for each elephant throughout the study period are shown in Table 3. Values from eight healthy calves with no clinical illness in the non-EEHV and prior-EEHV groups (i.e., all except E5) were averaged and analyzed across the three seasons in Thailand and within EEHV status with a comparison between the current study and published reference range values (Table 4). Packed-cell volume (PCV), red blood cell count (RBC), monocytes, and monocyte/heterophil (M:H) ratios were highest in the winter compared to the summer and the rainy season. There were no significant differences among the non-EEHV and prior-EEHV elephant groups (Figure 1 and Figure 2, and Appendix A). In addition, average hematology values of E5 in the 7 months before developing EEHV-HD were in the normal range and did not differ from the other elephants, except for PCV, which was lower compared to the study mean (Figure 3). In E5, marked decreases in WBC counts, lymphocytes, monocytes, and platelets were observed on the first day clinical signs of EEHV-HD appeared and continued until death on Day 3. The lowest PCV and RBC were found on the first day of clinical presentation (1.3 and 1.1 times the study mean, respectively). The lowest monocyte count and M:H ratio was found on the second day of clinical presentation (4.9 and 9.6 times, respectively). The lowest lymphocyte and platelet counts were found on the third day of clinical presentation (5.8 and 6.2 times, respectively). The viral load was high on the first day (6,902,328 vgc/mL) and decreased only slightly during treatment on Day 1 and 2 but surged again on Day 3 (Table 2).

### 3.3. Stress Indicator Concentrations

Overall mean concentrations of glucocorticoids and IgA for individuals during the study are shown in Table 5. Salivary cortisol concentrations were higher in the winter than the rainy and the summer, whereas FGM concentrations were higher in the summer than the rainy and the winter. SIgA concentrations in the rainy season were higher than those in the winter and the summer, whereas FIgA concentrations in the summer were higher than those in the winter and the rainy seasons (Figure 4). Salivary cortisol in the non-EEHV was lower than the prior EEHV groups significantly different. There were no differences in FGM, SIgA, or FIgA concentrations between the non-EEHV and prior-EEHV groups (Figure 5). In E5, fluctuations in SC, FGM, SIgA, and FIgA were observed during the 7 months of sample collection, with no significant differences compared to the normal range (Figure 6). However, in June 2019, the last month of sample collection, concentrations of SC, FGM, SIgA, and FIgA were all lower than the study mean and also the overall mean for that individual (Table 2).

### 3.4. Viremia Monitoring

Over the 12-month period, viremia was detected in two elephants (E3 and E5). For E3, low viremia (137 vgc/mL) was observed only once in the first month of sample collection without clinical signs of EEHV-HD, whereas a high viral load (6,902,328 vgc/mL) and EEHV-HD clinical signs were detected in E5 in the seventh month of sample collection (Table 2). Other elephants in the non-EEHV infection (E2, E8, and E9) and prior-EEHV (E1, E4, E6, and E7) groups did not show any clinical illness or EEHV viremia during the study.

## 4. Discussion

This is the first study to monitor longitudinal hematology and stress biomarkers (glucocorticoids and IgA) in saliva and feces in relation to EEHV status in Asian elephant calves. Results document normal range values of hematology and stress biomarkers in elephant calves, the age most at risk for developing EEHV-HD [5,16,17,51]. These normal range values can have predictive value for rapid diagnosis of early EEHV infection and improving survival outcomes [2]. In the one calf that was re-infected and died of HD, there were expected changes in hematology, but also decreases in stress-response measures, preliminarily suggesting they may also be informative and deserve further attention.

There was a seasonal effect on hematology values (PCV, RBC, monocytes, and M:H ratio) and welfare indicators in this study. Higher PCV and RBC have been observed in humans [52,53] and horses [54] during the winter in association with dehydration, although there was no indication that the healthy calves in this study experienced a lack of water. Monocytes and the M:H ratio were higher in the winter compared to the other two seasons. In humans, high WBC and monocytes in the winter could be a defense mechanism against infectious diseases [55] or associated with cold stress [56]. By contrast, in cattle, WBC and monocytes were higher in the summer season due to the release of corticosteroids or epinephrine as a result of relatively higher temperatures, which in turn increased leukocyte counts [57,58]. More work is needed to understand these seasonal effects and how they relate to health status.

Several studies have examined FGM and SIgA as welfare indicators to understand how tourist activities affect the quality of life of elephants in Thailand [34,42,44,59]. In this study, these techniques were used to determine how stress responses might be associated with compromised host immunity and the consequent development of EEHV-HD in young calves. However, only two elephants exhibited any viremia during the study. In one case, the viral load was low and not indicative of HD, and there were no changes in stress biomarker concentrations. The other elephant did develop HD and died within a few days. Concentrations of all biomarkers were markedly lower in the fecal and saliva samples collected 12 days before viremia was detected. No fecal or salivary samples were collected closer to or during the time clinical signs were observed, so the degree of these changes might have been even greater. A form of hypoadrenalism might have been at play in this calf, similar to a study in zoo black rhinoceros where reduced concentrations of FGM were found in association with the manifestation of severe skin disease [60]. As an important immune biomarker related to health status in humans and animals, decreases in IgA concentrations also have been related to chronic disease [46]. In dogs, significantly decreased concentrations of FIgA were found to be associated with inflammatory bowel disease [61]. In elephants, salivary IgA concentrations are lower in the summer, the season with the highest prevalence of gastrointestinal tract illnesses [42]. In E5, intermittent mild diarrhea was observed along with other EEHV clinical signs, so more studies to assess changing patterns of glucocorticoids and IgA associated with EEHV-HD cases are warranted.

Seasonal effects on stress indicators were found in this study, although not always in the same direction. A previous study of tourist elephants found salivary cortisol concentrations varied across environmental seasons, being high in the winter, intermediate in the summer, and lowest in the rainy season [42], similar to this study. In other species, such as pigs [62] and red deer [63], circulating glucocorticoid concentrations increased in response to cold stress. Additionally, the winter is the high tourist season in Thailand, when elephants have more activities [59,64] and are fed higher energy foods by tourists [42]. Therefore, high energy intake and energy demand for maintaining body temperature in cooler months might increase cortisol in blood circulation [64] followed by increases in saliva as noted in this study. Other studies in Asian elephants found FGM concentrations were highest in the winter in Spain [33] and during the logging season (June-August) in semi-captive elephants in Myanmar [65], while there were no seasonal differences in FGM concentrations in zoo elephants in the U.S. [66]. In this study, the highest FGM concentrations were found in the summer and so differed from our previous studies in adult elephants [42,59,64]. Calves often interact with tourists, but in this age group, the high tourist season was not associated with increases in adrenal activity. Temperatures are high during the summer in the northern part of Thailand (range, 36–42 °C) (www.tmd.go.th, accessed on 12 July 2020) and so might be related to physiological responses to thermal stress similar to that described for cattle [67,68] and sheep [69]. Moreover, variation in FGM concentrations can result from factors such as gender, diet, age, reproductive status, and habitat [70], and so should be considered in further studies of young calves.

In the previous study, SIgA concentrations were highest in the winter and high tourist season in adult elephants [42]. In this study, higher SIgA was found in the rainy season, which might be related to health status. In the rainy season, elephants generally have access to better quality food, which might result in good body condition scores and immunity. Future studies should monitor elephant body condition and health parameters in addition to stress biomarkers in older and younger age categories. The FIgA results also varied between young and adult elephants. In the previous study, FIgA excretion was consistent throughout the year in adult elephants [42], unlike this study where the highest FIgA concentrations were observed in the summer in young calves. Variations in FIgA concentration have been associated with factors such as a stressor (decrease), an acute illness (increase), and positive stimuli (increase) [46]. Thus, interpretation of FIgA as a stress biomarker needs to be done in consideration of other factors such as age, health status, and management in individual animals.

These findings confirm that animals that survive EEHV1A infection can be re-infected with the same viral subtype. The calf was viremic a year before the study and died soon after developing clinical signs during re-infection. One other study reported EEHV re-infection, but that involved two different types—EEHV1B followed by EEHV4 [11]. This phenomenon could be explained by the reactivation of latent viruses from the first infection [11]. In this calf, the severity of clinical signs in the primary infection was categorized as Level 4 [5], while the secondary infection was Level 5, the highest category indicating a serious infection [5]. Differences in severity of clinical signs might be prognostic of survival outcomes, but there is little empirical evidence of that to date.

Changes in hematology values, particularly WBC parameters, were associated with EEHV1A viremia in the calf that developed HD and died. The calf displayed lymphocytopenia, monocytopenia, and thrombocytopenia, which have been described in other reports [2,4,11,12]. Generally, hematology profiles are an important indicator of physiological changes in animals [71,72]. Monitoring the M:H ratio has been suggested to be another indicator of clinical EEHV progression, with a decrease associated with disease [2]. In the calf that died, the average M:H was low during the presentation of clinical signs compared with the study mean in this and another study [2]. Moreover, PVC, RBC, lymphocytes, monocytes, platelets, and the M:H ratio also were lower than normal, which is similar to other studies [2,4,11,12] and thus appear to be an important factor to monitor. Glucocorticoid and IgA concentrations in this calf in the 12 days prior to viremia also were markedly lower, about a half or more of those observed in the months prior to viremia, and in comparison, to the other elephants. While these data are very preliminary (*n* = 1), in addition to hematology, it might also be informative to assess stress-related biomarkers in addition to molecular-based PCR analysis to monitor EEHV status, allowing more timely treatment to improve survival outcomes.

The limitations of this study were the lack of EEHV-HD cases (only one) and the limited frequency of sampling to observe changes in the patterns of glucocorticoids or IgA in association with viremia during clinical signs. In clinical cases, saliva, feces, and blood should be collected daily to evaluate the correlation and consistency of all parameters when either clinical signs or viremia is detected. Longitudinal saliva and fecal sampling in more calves before, during, and after clinical EEHV presentation are required to fully understand any associations between stress and EEHV-HD.

## 5. Conclusions

This is the first longitudinal study that compared hematology values between non-EEHV and prior-EEHV infected groups and confirmed in the one calf that developed HD and died that changes in hematology parameters such as PCV, lymphocytes, monocytes, and platelets are indicators of clinical EEHV and a useful diagnostic tool before molecular confirmation. Analyses of several stress biomarkers revealed no differences in calves that had tested positive for EEHV-HD but survived compared to healthy controls. However, all biomarkers showed a marked decrease in samples collected 12 days prior to clinical signs in one calf that developed HD and died within 72 h. Although those data are very preliminary with an *n* = 1, analyses of stress factors might also provide early warning signs of EEHV viremia and so warrant further investigation. Through the longitudinal study, we found season had an effect on the concentrations of these indicators and so it is important to consider that when interpreting results in individual elephants.

## Figures and Tables

**Figure 1 animals-11-02530-f001:**
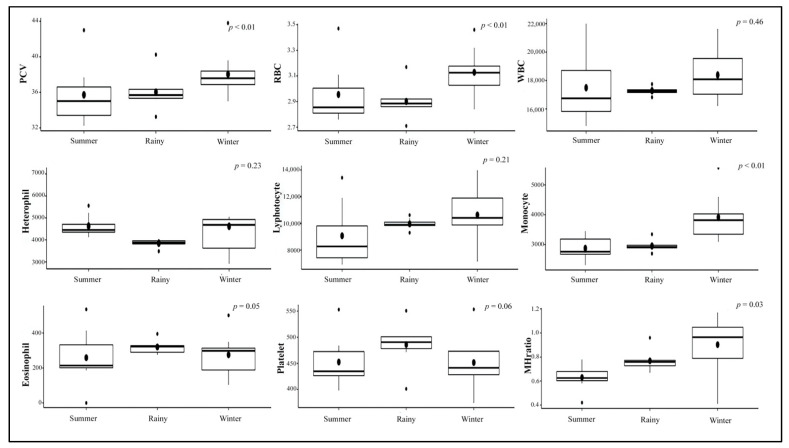
Boxplots of hematology values across three seasons in Thailand, excluding E5 that developed EEHV-HD and died during the study.

**Figure 2 animals-11-02530-f002:**
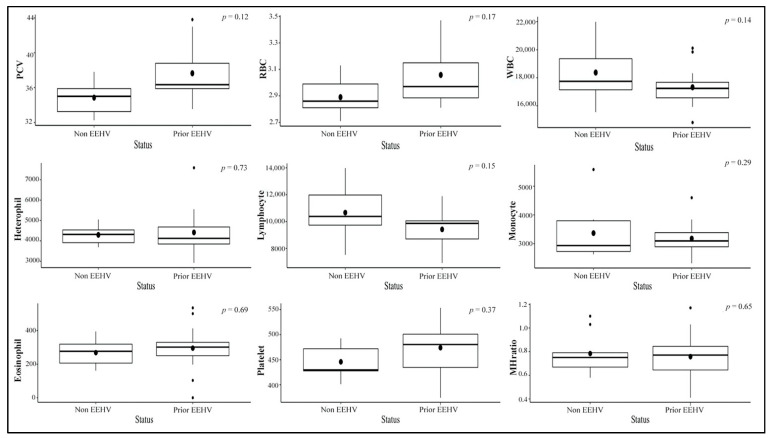
Boxplots of hematology values according to EEHV status (non-EEHV or prior-EEHV), excluding E5 that developed EEHV-HD and died during the study.

**Figure 3 animals-11-02530-f003:**
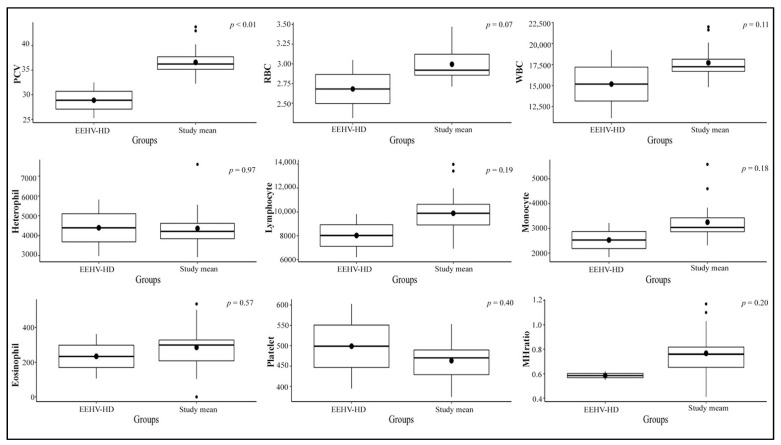
Boxplot presented significant differences in hematology between EEHV-HD and the study mean.

**Figure 4 animals-11-02530-f004:**
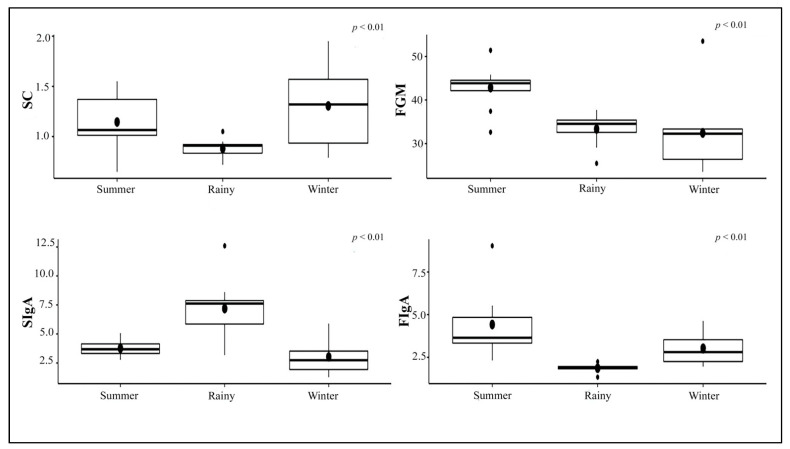
Boxplot presented concentrations of salivary cortisol (SC), fecal glucocorticoid metabolites (FGM), salivary IgA (SIgA), and fecal IgA (FIgA) categorized by season.

**Figure 5 animals-11-02530-f005:**
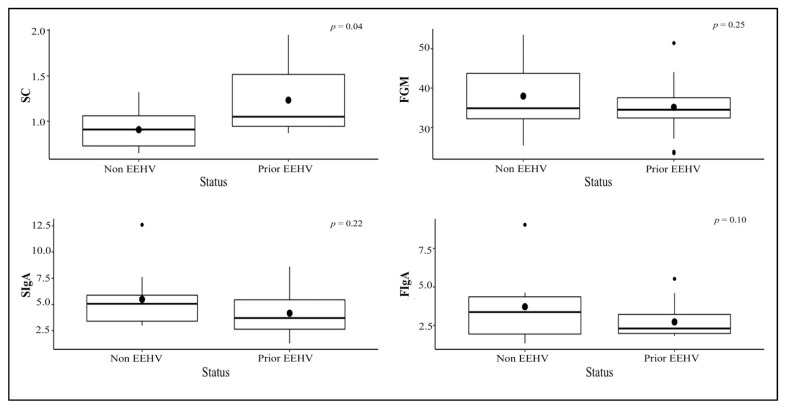
Boxplot presented concentrations of salivary cortisol (SC), fecal glucocorticoid metabolites (FGM), salivary IgA (SIgA), and fecal IgA (FIgA) categorized by EEHV status (non-EEHV or prior-EEHV), excluding E5 that developed EEHV-HD and died during the study.

**Figure 6 animals-11-02530-f006:**
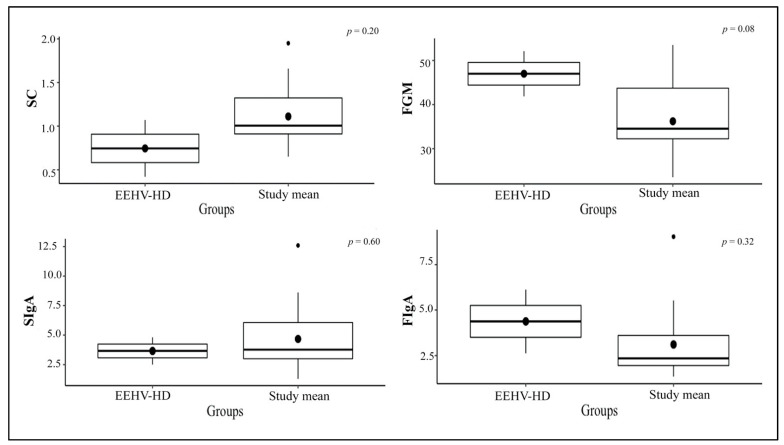
Boxplot presented significant differences in biological indicators between EEHV-HD and study mean.

**Table 1 animals-11-02530-t001:** Information on the elephant calves in the study and the EEHV history.

ID	Camp	Gender	Weaning Age (Months)	Age at Start of Study (Months)	EEHV Category	Detection of EEHV				EEHV Subtype	Clinical Signs and Treatment
						Date	Age (Months)	Test	Highest Viral Load (vgc/mL)		
E1	A	Female	36	77	Prior-EEHV ^1^	July 2017	60	Conventional PCR	negative	4	Signs: Depression and diarrhea
											Tx: Acyclovir, fluids, and supportive therapies ^4^
E2	B	Male	30	60	Non-EEHV ^2^	N/A				N/A	N/A
E3	C	Female	30	39	Prior-EEHV	October 2018	36	Real-time PCR	1,152,837	1A	Signs: Lethargy, depression, and facial edema
											Tx: Acyclovir, fluids, and supportive therapies ^4^
E4	D	Female	15	35	Prior-EEHV	July 2018	28	Real-time PCR	25,607	4	Signs: Depressing and diarrhea
											Tx: Acyclovir, fluids, and supportive therapies ^4^
E5	E	Male	12	20	Prior-EEHV ^3^	June 2018 and June 2019	14 and 26	Real-time PCR	13 and 6,902,328	1A	1st infection
											Signs: Lethargy, depression, and facial edema
											Tx: Famciclovir, fluids, and supportive therapies
											2nd infection
											Signs: Lethargy, depression, fever, and facial edema, diarrhea
											Tx: Acyclovir, fluids, and supportive therapies ^4^
E6	A	Female	35	78	Prior-EEHV	July 2018	72	Conventional PCR	negative	4	Signs: Depression and diarrhea
											Tx: Acyclovir, fluids, and supportive therapies ^4^
E7	F	Male	12	24	Prior-EEHV	July 2018	18	Real-time PCR	1,306,864	4	Signs: Depression and diarrhea
											Tx: Acyclovir, fluids, and supportive therapies ^4^
E8	B	Male	34	60	Non-EEHV	N/A				N/A	N/A
E9	B	Male	34	48	Non-EEHV	N/A				N/A	N/A

^1^ Survived EEHV-HD prior to the study; ^2^ No history of EEHV-HD and PCR negative in the month prior to study onset; ^3^ Re-infected with EEHV-HD during the study and died; ^4^ Supportive therapies were based on guidelines for EEHV management in Asia (Luz and Howard, 2016) and the United Kingdom (Cracknell, 2008); N/A, not applicable.

**Table 2 animals-11-02530-t002:** Summary hematology values change, and stress indicators result in an elephant (E5) that developed clinical signs of EEHV-HD and died during the study period.

Parameters	Study Means	18 December 2018	9 January 2019	8 February 2019	8 March 2019	6 April 2019	6 May 2019	4 June 2019	16 June 2019	17 June 2019	18 June 2019
Viral load (vgc/mL)	-	Undetected	Undetected	Undetected	Undetected	Undetected	Undetected	Undetected	6,902,328	4,845,925	5,744,321
PCV (%)	37.3 ± 0.38	30	31	36	33	36	33	32	29	30	31
RBC (×10 ^6^ cells/μL)	3.10 ± 0.03	2.81	2.93	3.37	3.08	3.33	3.02	2.88	2.73	3.29	2.95
WBC (cells/μL)	18825 ± 351	16,230	17,760	20,640	22,300	15,370	15,010	14,140	10,110	14,300	8330
Heterophils (cells/μL)	4247 ± 136	3246	2309	7018	10,704	4765	3753	3394	5561	8065	5081
Lymphocytes (cells/μL)	10,675 ± 358	10,063	12,077	8050	9143	9222	8256	7353	2325	2054	1833
Monocytes (cells/μL)	3459 ± 219	2434	2842	5366	2230	1383	3002	2969	2224	700	1333
Eosinophile (cells/μL)	296 ± 28	487	532	206	618	0	0	428	143	0	83
Platelets (×10 ^3^ cells/μL)	458 ± 8	515	553	802	541	459	580	540	194	98	74
Monocyte/Heterophil ratio	0.87 ± 0.05	0.75	1.23	0.76	0.21	0.29	0.80	0.87	0.40	0.09	0.26
Salivary cortisol (ng/mL)	1.17 ± 0.08	1.03	0.91	1.57	0.78	1.27	0.40	0.39	No sample	No sample	No sample
FGM (ng/g)	36.1 ± 3.19	72.30	62.59	34.37	39.26	103.11	48.73	15.56	No sample	No sample	No sample
SIgA (µg/mL)	3.53 ± 0.31	3.70	4.98	5.50	5.04	5.49	2.78	1.70	No sample	No sample	No sample
FIgA (µg/mL)	1.85 ± 0.10	2.42	2.57	1.41	1.55	12.22	2.57	0.94	No sample	No sample	No sample

**Table 3 animals-11-02530-t003:** Mean (±SEM) hematology values of individual elephants in the study.

Parameter	Elephant ID (Blood Sample Number)									Range
	E1 (*n* = 13)	E2 (*n* = 13)	E3 (*n* = 13)	E4 (*n* = 13)	E5 * (*n* = 7)	E6 (*n* = 13)	E7 (*n* = 13)	E8 (*n* = 13)	E9 (*n* = 13)	
PCV (%)	42.40 ± 1.08	33.50 ± 0.80	37.60 ± 1.18	36.00 ± 0.30	28.90 ± 3.62	36.90 ± 0.57	35.40 ± 1.11	35.70 ± 1.03	35.40 ± 1.38	32.25–43.80
RBC (×10 ^6^ cells/μL)	3.37 ± 0.08	2.84 ± 0.08	3.10 ± 0.12	2.88 ± 0.02	2.68 ± 0.37	3.00 ± 0.06	2.93 ± 0.99	2.95 ± 0.09	2.89 ± 0.08	2.17–3.47
WBC (cells/μL)	17,731 ± 1245	19,700 ± 1236	17,979 ± 942	16,124 ± 709	15,181 ± 4051	17,262 ± 611	17,523 ± 489	17,165 ± 910	18,257 ± 3088	14,810–21,990
Heterophils (cells/μL)	5147 ± 1246	4467 ± 303	3687 ± 384	4629 ± 500	4300 ± 1421	4176 ± 539	4421 ± 274	4421 ± 282	3989 ± 274	2920–7587
Lymphocyte (cells/μL)	10,051 ± 1033	11,928 ± 877	10,633 ± 751	7982 ± 956	8020 ± 181	8725 ± 813	9698 ± 301	9367 ± 569	10,706 ± 1865	6918–13,980
Monocytes (cells/μL)	3059 ± 192	3817 ± 892	3176 ± 153	3298 ± 678	2528 ± 690	3175 ± 143	3176 ± 333	3167 ± 342	3111 ± 352	2316–5578
Eosinophils (cells/μL)	374 ± 80	321 ± 53	265 ± 33	143 ± 96	2234 ± 128	413 ± 50	282 ± 37	276 ± 35	208 ± 35	0–535
Platelets (×10 ^3^ cells/μL)	552 ± 23	465 ± 14	464 ± 18	463 ± 20	499 ± 104	424 ± 39	466 ± 15	465 ± 18	416 ± 8	374–554
M:H ratio	0.66 ± 0.16	0.84 ± 0.13	0.90 ± 0.14	0.74 ± 0.17	0.58 ± 0.03	0.77 ± 0.07	0.72 ± 0.05	0.74 ± 0.05	0.78 ± 0.13	0.41–1.17

* Died before the end of sample collection.

**Table 4 animals-11-02530-t004:** Mean (±SEM) hematology values and stress indicator concentrations in eight young calves without clinical illness throughout the study period, excluding E5 that developed EEHV-HD and died during the study.

Variables	Mean Value in the Study (*n* = 104)	Range	Fowler et al. (2006)	Janyamethakul et al. (2017)	
				Male	Female
PCV (%)	37.3 ± 0.38	32.25–43.80	30–40	29.40–40.70	27.80–43.00
RBC (×10 ^6^ cells/μL)	3.10 ± 0.03	2.17–3.47	2.00–5.00	1.90–3.20	1.90–3.10
WBC (cells/μL)	18,825 ± 351	14,810–21,990	10,000–18,000	7924–21,890	7202.50–23,220.50
Heterophils (cells/μL)	4247 ± 136	2920–7587	2000–4000	967–13426	828.70–13,514.30
Lymphocytes (cells/μL)	10,675 ± 358	6918–13,980	5000–8000	1672–11,179	1064.10–12,032.80
Monocytes (cells/μL)	3459 ± 219	2316–5578	2000–4000	0–2391	0–3298
Eosinophils (cells/μL)	296 ± 28	0–535	100–1000	0–867	0–1170
Platelets (×10 ^3^ cells/μL)	458 ± 8.38	374–554	200–600	102–578	105.30–598.70
Monocyte/Heterophil ratio	0.87 ± 0.05	0.41–1.17	1.18–3.57	ND	ND
Salivary cortisol (ng/mL)	1.17 ± 0.08	0.65–1.95	ND	ND	ND
FGM (ng/mL)	36.1 ± 3.19	23.55–53.51	ND	ND	ND
SIgA (µg/mL)	3.53 ± 0.31	1.28–12.60	ND	ND	ND
FIgA (µg/mL)	1.85 ± 0.10	1.33 9.04	ND	ND	ND

ND, no data.

**Table 5 animals-11-02530-t005:** Overall mean (±SEM) concentrations of salivary cortisol, fecal glucocorticoid metabolites (FGM), salivary IgA (SIgA), and fecal IgA (FIgA) in individual elephants in the study.

Variable	Elephant ID									Range
	E1 (*n =* 12)	E2 (*n =* 12)	E3 (*n =* 12)	E4 (*n =* 12)	E5 * (*n =* 7)	E6 (*n =* 12)	E7 (*n =* 12)	E8 (*n =* 12)	E9 (*n =* 12)	
Salivary cortisol (ng/mL)	1.40 ± 0.18	0.72 ± 0.04	0.97 ± 0.02	1.40 ± 0.30	0.75 ± 0.32	1.33 ± 0.21	1.06 ± 0.13	1.10 ± 0.12	0.90 ± 0.01	0.65–1.95
FGM (ng/mL)	31.3 ± 4.14	36.10 ± 5.02	36.00 ± 4.85	39.8 ± 5.83	47.00 ± 5.14	32.00 ± 4.10	36.70 ± 3.72	37.00 ± 3.47	40.90 ± 8.21	23.55–53.51
SIgA (µg/mL)	2.63 ± 0.66	7.85 ± 293	4.66 ± 1.18	4.328 ± 1.70	3.66 ± 1.15	4.33 ± 184	5.14 ± 1.75	4.72 ± 1.45	3.91 ± 0.61	1.28–12.60
FIgA (µg/mL)	2.43 ± 0.38	5.00 ± 2.23	3.38 ± 1.08	2.46 ± 0.55	4.38 ± 1.75	2.16 ± 0.15	3.23 ± 0.80	2.97 ± 0.53	3.18 ± 0.74	1.33–9.04

* Developed EEHV-HD and died before the end of sample collection.

## Data Availability

The data presented in this study are available on request from corresponding author.

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
