# Peer review of "A Longitudinal Study of Hematology and Stress Biomarker Profiles in Young Asian Elephants (Elephas Maximus) in Relation to Elephant Endotheliotropic Herpesvirus (EEHV) in Thailand"

_animals, 2021, doi:10.3390/ani11092530_

Round 1

Reviewer 1 Report

In “Hematology, fecal and salivary glucocorticoids, and IgA measures assoc. with EEHV outcomes in young Asian elephants (Elephas maximus) in Thailand,” Boonprasert et. al. present important baseline data from 8 elephant calves. I am excluding the 9th calf (E5) due to its active infection and death near the midpoint of the study. This manuscript is well written and supported with good evidence.

Major comments:

1) The major drawback to this study is that no conclusions or statements can or should be made in affirmation of any supposed predictive value of changing values ahead of EEHV infection. Because there is just an N of 1 elephant that become infected (re-infected or re-activation) during the study there should be no statements or conclusions drawn from that n of 1. E5 can be, of course, looked at as a case study within this larger collection of baseline data, but statements made in the Abstract and Discussion related to conclusive evidence of predictive value must be removed.  Further, I do not believe that you can draw any conclusions about the limited information collected from E5 just prior to death, if you look at the lab values and measures, they were incredibly variable in this animal seemingly across all timepoints.

2) While collecting this baseline data is important, there is no inclusion of a conversation in the Introduction about how identifying these baseline measures may improve outcomes of EEHV in young elephants. Are there any possibilities or evidence of early diagnosis leading to improved outcomes in elephants? Other mammals? Opening up different therapy possibilities? Limiting spread of disease? This seems to be a major impact of collecting baseline data and could add weight to this manuscript.

Minor comments:

1) Please revise figures to align data according to time, changing from an x axis of “rainy…summer…winter” to summer…rainy…winter to make sure that the data flow temporally for the international reader. It may even be helpful that, within the figures, the months that define each season are included for the international reader that is unfamiliar with the seasons/weather of Thailand.

2) In Table 1, there is no mention of facial edema during the primary infection of E5 but it is mentioned in the discussion. Please clarify either the Table information or the discussion section. Is facial edema predictive of poor prognosis?

3) In the discussion, the authors presume that E5 may have had a preexisting bacterial infection prior to re-infection or re-activation of HD but provide no evidence to support that statement. This should be removed. The authors also mention “2 days prior” of diarrheal disease in E5 but the timeline is blurry because they have already asserted that symptom onset was clear 3 days prior to death. Please clarify the timeline of events and symptoms.

4) The discussions on stressors and seasonality were important and well presented.

Author Response

Dear reviewer

First of all, we would like to thank reviewers for their constructive comments on the content of this manuscript. We have amended the article as follow the reviewer' suggestion. For those changed details "Please see the attachment".

Best regards

Khajohnpat Boonprasert

Reviewer 2 Report

Line 31 - Immunoglobulin A

Line 36 – Virulence is the ability of a microorganism to cause disease. So, do not exist There are no high virulent or low virulent diseases.

Line 37 “young…calves”, calves are young, so remove young or use a specific age, for example “4 months..calves”

Line 48 “HD”, the first time, write in full

Line 57 “Elephant endotheliotropic herpesvirus (EEHV)”

Line 62 “has been shown to be” change to “are”

Line 109 “license number S2/2561”

Line 145 “2,000 RPM for 2 minutes at 15oC”, use G and not RPM (alternatively or include information about the centrifuge, namely the axis radius)

Line 145 and elsewhere “oC” may be changed to “ºC”

Line 164 and elsewhere “Macherey-Nagel, Germany”, include the sequence (Company, City, Country).

Line 177 and elsewhere “0.5 X” should be 0.5×

Table 2 and elsewhere “vgc/ml” may be “vgc/mL”

Table 2 and elsewhere “/ul” may be “μL”

Line 406 “could in fact”, remove in fact

Author Response

Dear reviewer 

First of all, we would like to thank the reviewer for their constructive comments on the content of this manuscript. We have amended the article as follow the reviewer' suggestion. For those changed details "Please see the attachment".

Best regards

khajohnpat Boonprasert

Reviewer 3 Report

This study documents hematology and stress biomarkers over time in Asian elephants. It is a very interesting and needed study, because it focuses on elephants at the age when EEHV HD is commonly diagnosed. Documenting the normal range values in this age group is important for EEHV monitoring. One elephant developed EEHV HD during the study, which confirmed that these markers change as disease progresses. The identification of decreased stress biomarkers 12 weeks before EEHV viremia is interesting. Since only one elephant developed EEHV HD, the changes in these biomarkers will need to be confirmed in additional studies. 

I think the title of the article is misleading. It implies that the study is associating these markers with outcomes, which cannot be done with only one elephant developing disease. The article is really characterizing these values over time in young asian elephants, I don't see enough evidence to say that it's looking at an association with outcomes.

Other than that, it's a very good study that should be published.

Author Response

Dear reviewer 

First of all, we would like to thank the reviewer for their constructive comments on the content of this manuscript. We have amended the article as follow reviewer suggestion. For those changed details "Please see the attachment".

Best regards

Khajohnpat Boonprasert
